# Learning to Perform Local Rewriting for Combinatorial Optimization

**Xinyun Chen** [*]
UC Berkeley
xinyun.chen@berkeley.edu

**Yuandong Tian**
Facebook AI Research
yuandong@fb.com

## Abstract

Search-based methods for hard combinatorial optimization are often guided by heuristics. Tuning heuristics in various conditions and situations is often time-consuming. In this paper, we propose `NeuRewriter` that learns a policy to pick heuristics and rewrite the local components of the current solution to iteratively improve it until convergence. The policy factorizes into a region-picking and a rule-picking component, each parameterized by a neural network trained with actor-critic methods in reinforcement learning. `NeuRewriter` captures the general structure of combinatorial problems and shows strong performance in three versatile tasks: expression simplification, online job scheduling and vehicle routing problems. `NeuRewriter` outperforms the expression simplification component in Z3 [15]; outperforms DeepRM [33] and Google OR-tools [19] in online job scheduling; and outperforms recent neural baselines [35, 29] and Google OR-tools [19] in vehicle routing problems. [2]

## 1 Introduction

Solving combinatorial problems is a long-standing challenge and has a lot of practical applications (e.g., job scheduling, theorem proving, planning, decision making). While problems with specific structures (e.g., shortest path) can be solved efficiently with proven algorithms (e.g, dynamic programming, greedy approach, search), many combinatorial problems are NP-hard and rely on manually designed heuristics to improve the quality of solutions [1, 40, 27].

Although it is usually easy to come up with many heuristics, determining when and where such heuristics should be applied, and how they should be prioritized, is time-consuming. It takes commercial solvers decades to tune to strong performance in practical problems [15, 44, 19].

To address this issue, previous works use neural networks to predict a complete solution from scratch, given a complete description of the problem [50, 33, 29, 21]. While this avoids search and tuning, a direct prediction could be difficult when the number of variables grows.

Improving iteratively from an existing solution is a common approach for continuous solution spaces, e.g, trajectory optimization in robotics [34, 47, 31]. However, such methods relying on gradient information to guide the search, is not applicable for discrete solution spaces due to indifferentiablity.

To address this problem, we directly learn a neural-based policy that improves the current solution by iteratively rewriting a local part of it until convergence. Inspired by the problem structures, the policy is factorized into two parts: the region-picking and the rule-picking policy, and is trained end-to-end with reinforcement learning, rewarding cumulative improvement of the solution.

We apply our approach, `NeuRewriter`, to three different domains: expression simplification, online job scheduling, and vehicle routing problems. We show that `NeuRewriter` is better than strong

---

[*]Work partially done when interning at Facebook AI Research.

[2]The code is available at `https://github.com/facebookresearch/neural-rewriter`.

heuristics using multiple metrics. For expression simplification, `NeuRewriter` outperforms the expression simplification component in Z3 [15]. For online job scheduling, under a controlled setting, `NeuRewriter` outperforms Google OR-tools [19] in terms of both speed and quality of the solution, and DeepRM [33], a neural-based approach that predicts a holistic scheduling plan, by large margins especially in more complicated setting (e.g., with more heterogeneous resources). For vehicle routing problems, `NeuRewriter` outperforms two recent neural network approaches [35, 29] and Google OR-tools [19]. Furthermore, extensive ablation studies show that our approach works well in different situations (e.g., different expression lengths, non-uniform job/resource distribution), and transfers well when distribution shifts (e.g., test on longer expressions than those used for training).

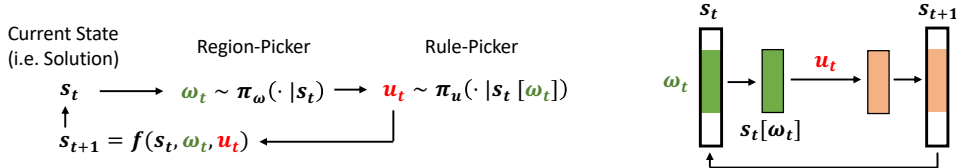

Figure 1: The framework of our neural rewriter. Given the current state (i.e., solution to the optimization problem) $s_t$, we first pick a region $\omega_t$ by the region-picking policy $\pi_\omega(\omega_t|s_t)$, and then pick a rewriting rule $u_t$ using the rule-picking policy $\pi_u(u_t|s_t[\omega_t])$, where $\pi_u(u_t|s_t[\omega_t])$ gives the probability distribution of applying each rewriting rule $u \in \mathcal{U}$ to the partial solution. Once the partial solution is updated, we obtain an improved solution $s_{t+1}$ and repeat the process until convergence.

## 2 Related Work

**Methods**. Using neural network models for combinatorial optimization has been explored in the last few years. A straightforward idea is to construct a solution directly (e.g., with a Seq2Seq model) from the problem specification [50, 6, 33, 28]. However, such approaches might meet with difficulties if the problem has complex configurations, as our evaluation indicates. In contrast, our paper focuses on iterative improvement of a *complete* solution.

Trajectory optimization with local gradient information has been widely studied in robotics with many effective techniques [34, 9, 51, 47, 32, 31]. For discrete problems, it is possible to apply continuous relaxation and apply gradient descent [10]. In contrast, we *learn the gradient* from previous experience to optimize a complete solution, similar to data-driven descent [49] and synthetic gradient [26].

At a high level, our framework is closely connected with the local search pipeline. Specifically, we can leverage our learned RL policy to guide the local search, i.e., to decide which neighbor solution to move to. We will demonstrate that in our evaluated tasks, our approach outperforms several local search algorithms guided by manually designed heuristics, and softwares supporting more advanced local search algorithms, i.e., Z3 [15] and OR-tools [19].

**Applications**. For expression simplification, some recent work use deep neural networks to discover equivalent expressions [11, 2, 52]. In particular, [11] trains a deep neural network to rewrite algebraic expressions with supervised learning, which requires a collection of ground truth rewriting paths, and may not find novel rewriting routines. We mitigate these limitations using reinforcement learning.

Job scheduling and resource management problems are ubiquitous and fundamental in computer systems. Various work have studied these problems from both theoretical and empirical sides [8, 20, 3, 42, 48, 33, 13]. In particular, a recent line of work studies deep reinforcement learning for job scheduling [33, 13] and vehicle routing problems [29, 35].

Our approach is tested on multiple domains with extensive ablation studies, and could also be extended to other closely related tasks such as code optimization [41, 12], theorem proving [25, 30, 4, 24], text simplification [14, 37, 18], and classical combinatorial optimization problems beyond routing problems [16, 28, 7, 50, 27], e.g., Vertex Cover Problem [5].

## 3 Problem Setup

Let $\mathcal{S}$ be the space of all feasible solutions in the problem domain, and $c : \mathcal{S} \to \mathbb{R}$ be the cost function. The goal of optimization is to find $\arg\min_{s \in \mathcal{S}} c(s)$. In this work, instead of finding a solution from scratch, we first construct a feasible one, then make incremental improvement by iteratively applying *local rewriting rules* to the existing solution until convergence. Our rewriting formulation is especially suitable for problems with the following properties: **(1)** a feasible solution

is easy to find; **(2)** the search space has well-behaved local structures, which could be utilized to incrementally improve the solution. For such problems, a complete solution provides a full context for the improvement using a rewriting-based approach, allowing additional features to be computed, which is hard to obtain if the solution is generated from scratch; meanwhile, different solutions might share a common routine towards the optimum, which could be represented as local rewriting rules. For example, it is much easier to decide whether to postpone jobs with large resource requirements when an existing job schedule is provided. Furthermore, simple rules like swapping two jobs could improve the performance.

Formally, each solution is a *state*, and each local region and the associated rewriting rule is an *action*.

**Optimization as a rewriting problem.** Let $\mathcal{U}$ be the rewriting ruleset. Suppose $s_t$ is the current solution (or state) at iteration $t$. We first compute a state-dependent *region set* $\Omega(s_t)$, then pick a region $\omega_t \in \Omega(s_t)$ using the *region-picking policy* $\pi_\omega(\omega_t|s_t)$. We then pick a rewriting rule $u_t$ applicable to that region $\omega_t$ using the *rule-picking policy* $\pi_u(u_t|s_t[\omega_t])$, where $s_t[\omega_t]$ is a subset of state $s_t$. We then apply this rewriting rule $u_t \in \mathcal{U}$ to $s_t[\omega_t]$, and obtain the next state $s_{t+1} = f(s_t, \omega_t, u_t)$. Given an initial solution (or state) $s_0$, our goal is to find a sequence of rewriting steps $(s_0, (\omega_0, u_0)), (s_1, (\omega_1, u_1)), ..., (s_{T-1}, (\omega_{T-1}, u_{T-1})), s_T$ so that the final cost $c(s_T)$ is minimized.

To tackle a rewriting problem, rule-based rewriters with manually-designed rewriting routines have been proposed [23]. However, manually designing such routines is not a trivial task. An incomplete set of routines often leads to an inefficient exhaustive search, while a set of kaleidoscopic routines is often cumbersome to design, hard to maintain and lacks flexibility.

In this paper, we propose to train a neural network instead, using reinforcement learning. Recent advance in deep reinforcement learning suggests the potential of well-trained models to discover novel effective policies, such as demonstrated in Computer Go [43] and video games [36]. Moreover, by leveraging reinforcement learning, our approach could be extended to a broader range of problems that could be hard for rule-based rewriters and classic search algorithms. For example, we can design the reward to take the validity of the solution into account, so that we can start with an infeasible solution and then move towards a feasible one. On the other hand, we can also train the neural network to explore the connections between different solutions in the search space. In our evaluation, we demonstrate that our approach (1) mitigates laborious human efforts, (2) discovers novel rewriting paths from its own exploration, and (3) finds better solution to optimization problem than the current state-of-the-art and traditional heuristic-based software packages tuned for decades.

## 4 Neural Rewriter Model

In the following, we present the design of our rewriting model, i.e., `NeuRewriter`. We first provide an overview of our model framework, then present the design details for different applications.

### 4.1 Model Overview

Figure 1 illustrates the overall framework of our neural rewriter, and we describe the two key components for rewriting as follows. More details can be found in Appendix C.

**Score predictor.** Given the state $s_t$, the score predictor computes a score $Q(s_t, \omega_t)$ for every $\omega_t \in \Omega(s_t)$, which measures the benefit of rewriting $s_t[\omega_t]$. A high score indicates that rewriting $s_t[\omega_t]$ could be desirable. Note that $\Omega(s_t)$ is a problem-dependent region set. For expression simplification, $\Omega(s_t)$ includes all sub-trees of the expression parse trees; for job scheduling, $\Omega(s_t)$ covers all job nodes for scheduling; and for vehicle routing, it includes all nodes in the route.

**Rule selector.** Given $s_t[\omega_t]$ to be rewritten, the rule-picking policy predicts a probability distribution $\pi_u(s_t[\omega_t])$ over the entire ruleset $\mathcal{U}$, and selects a rule $u_t \in \mathcal{U}$ to apply accordingly.

### 4.2 Training Details

Let $(s_0, (\omega_0, u_0)), ..., (s_{T-1}, (\omega_{T-1}, u_{T-1})), s_T$ be the rewriting sequence in the forward pass.

**Reward function**. We define $r(s_t, (\omega_t, u_t))$ as $r(s_t, (\omega_t, u_t)) = c(s_t) - c(s_{t+1})$, where $c(\cdot)$ is the task-specific cost function in Section 3.

**Q-Actor-Critic training.** We train the region-picking policy $\pi_\omega$ and rule-picking policy $\pi_u$ simultaneously. For $\pi_\omega(\omega_t|s_t; \theta)$, we parameterize it as a softmax of the underlying $Q(s_t, \omega_t; \theta)$ function:

$$\pi_\omega(\omega_t|s_t; \theta) = \frac{\exp(Q(s_t, \omega_t; \theta))}{\sum_{\omega_t} \exp(Q(s_t, \omega_t; \theta))} \tag{1}$$

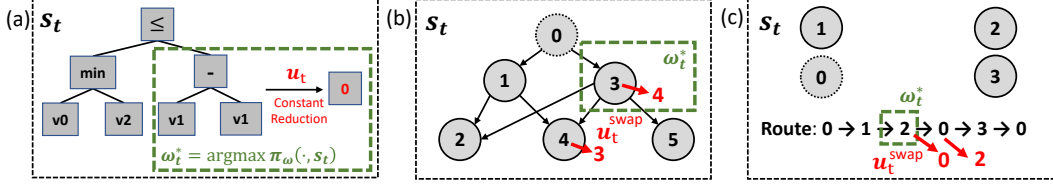

Figure 2: The instantiation of `NeuRewriter` for different domains: **(a)** expression simplification; **(b)** job scheduling; and **(c)** vehicle routing. In **(a)**, $s_t$ is the expression parse tree, where each square represents a node in the tree. The set $\Omega(s_t)$ includes every sub-tree rooted at a non-terminal node, from which the region-picking policy selects $\omega_t \sim \pi_\omega(\omega_t|s_t)$ to rewrite. Afterwards, the rule-picking policy predicts a rewriting rule $u_t \in \mathcal{U}$, then rewrites the sub-tree $\omega_t$ to get the new tree $s_{t+1}$. In **(b)**, $s_t$ is the dependency graph representation of the job schedule. Each circle with index greater than 0 represents a job node, and node 0 is an additional one representing the machine. Edges in the graph reflect job dependencies. The region-picking policy selects a job $\omega_t$ to re-schedule from all job nodes, then the rule-picking policy chooses a moving action $u_t$ for $\omega_t$, then modifies $s_t$ to get a new dependency graph $s_{t+1}$. In **(c)**, $s_t$ is the current route, and $\omega_t$ is the node selected to change the visit order. Node 0 is the depot, and other nodes are customers with certain resource demands. The region-picking policy and the rule-picking policy work similarly to the job scheduling ones.

and instead learn $Q(s_t, \omega_t; \theta)$ by fitting it to the cumulative reward sampled from the current policies $\pi_\omega$ and $\pi_u$:

$$L_\omega(\theta) = \frac{1}{T} \sum_{t=0}^{T-1} (\sum_{t'=t}^{T-1} \gamma^{t'-t} r(s'_t, (\omega'_t, u'_t)) - Q(s_t, \omega_t; \theta))^2 \qquad (2)$$

Where $T$ is the length of the episode (i.e., the number of rewriting steps), and $\gamma$ is the decay factor.

For rule-picking policy $\pi_u(u_t|s_t[\omega_t]; \phi)$, we employ the Advantage Actor-Critic algorithm [45] with the learned $Q(s_t, \omega_t; \theta)$ as the critic, and thus avoid boot-strapping which could cause sample insufficiency and instability in training. This formulation is similar in spirit to soft-Q learning [22]. Denoting $\Delta(s_t, (\omega_t, u_t)) \equiv \sum_{t'=t}^{T-1} \gamma^{t'-t} r(s'_t, (\omega'_t, u'_t)) - Q(s_t, \omega_t; \theta)$ as the advantage function, the loss function of the rule selector is:

$$L_u(\phi) = -\sum_{t=0}^{T-1} \Delta(s_t, (\omega_t, u_t)) \log \pi_u(u_t|s_t[\omega_t]; \phi) \qquad (3)$$

The overall loss function is $L(\theta, \phi) = L_u(\phi) + \alpha L_\omega(\theta)$, where $\alpha$ is a hyper-parameter. More training details can be found in Appendix D.

## 5 Applications

In the following sections, we discuss the application of our rewriting approach to three different domains: expression simplification, online job scheduling, and vehicle routing. In expression simplification, we minimize the expression length using a well-defined semantics-preserving rewriting ruleset. In online job scheduling, we aim to reduce the overall waiting time of jobs. In vehicle routing, we aim to minimize the total tour length.

### 5.1 Expression Simplification

We first apply our approach to expression simplification domain. In particular, we consider expressions in Halide, a domain-specific language for high-performance image processing [39], which is widely used at scale in multiple products of Google (e.g., YouTube) and Adobe Photoshop. Simplifying Halide expressions is an important step towards the optimization of the entire code. To this end, a rule-based rewriter is implemented for the expressions, which is carefully tuned with manually-designed heuristics. The grammar of the expressions considered in the rewriter is specified in Appendix A.1. Notice that the grammar includes a more comprehensive operator set than previous works on finding equivalent expressions, which consider only boolean expressions [2, 17] or a subset of algorithmic operations [2]. The rewriter includes hundreds of manually-designed rewriting templates. Given an expression, the rewriter checks the templates in a pre-designed order, and applies those rewriting templates that match any sub-expression of the input.

After investigating the rewriting templates in the rule-based rewriter, we find that a large number of rewriting templates enumerate specific cases for an *uphill rule*, which lengthens the expression first

and shortens it later (e.g., "min/max" expansion). Similar to momentum terms in gradient descent for continuous optimization, such rules are used to escape a local optimum. However, they should only be applied when the initial expression satisfies certain *pre-conditions*, which is traditionally specified by manual design, a cumbersome process that is hard to generalize.

Observing these limitations, we hypothesize that a neural network model has the potential of doing a better job than the rule-based rewriter. In particular, we propose to only keep the core rewriting rules in the ruleset, remove all unnecessary pre-conditions, and let the neural network decide which and when to apply each rewriting rule. In this way, the neural rewriter has a better flexibility than the rule-based rewriter, because it can learn such rewriting decisions from data, and has the ability of discovering novel rewriting patterns that are not included in the rule-based rewriter.

**Ruleset**. We incorporate two kinds of templates from Halide rewriting ruleset. The first kind is simple rules (e.g., $v - v \rightarrow 0$), while the second one is the uphill rules after removing their manually designed pre-conditions that do not affect the validity of the rewriting. In this way, a ruleset with $|\mathcal{U}| = 19$ categories is built. See Appendix B.1 for more details.

**Model specification.** We use expression parse trees as the input, and employ the N-ary Tree-LSTM designed in [46] as the input encoder to compute the embedding for each node in the tree. Both the score predictor and the rule selector are fully connected neural networks, taken the LSTM embeddings as the input. More details can be found in Appendix C.1.

## 5.2 Job Scheduling Problem

We also study the job scheduling problem, using the problem setup in [33].

**Notation**. Suppose we have a machine with $D$ types of resources. Each job $j$ is specified as $v_j = (\boldsymbol{\rho}_j, A_j, T_j)$, where the $D$-dimensional vector $\boldsymbol{\rho}_j = [\rho_{jd}]$ denotes the required portion $0 \leq \rho_{jd} \leq 1$ of the resource type $d$, $A_j$ is the arrival timestep, and $T_j$ is the duration. In addition, we define $B_j$ as the scheduled beginning time, and $C_j = B_j + T_j$ as the completion time.

We assume that the resource requirement is fixed during the entire job execution, each job must run continuously until finishing, and no preemption is allowed. We adopt an online setting: there is a pending job queue that can hold at most $W$ jobs. When a new job arrives, it can either be allocated immediately, or be added to the queue. If the queue is already full, to make space for the new job, at least one job in the queue needs to be scheduled immediately. The goal is to find a time schedule for every job, so that the average waiting time is as short as possible.

**Ruleset**. The set of rewriting rules is to re-schedule a job $v_j$ and allocate it after another job $v_{j'}$ finishes or at its arrival time $A_j$. See Appendix B.2 for details of a rewriting step. The size of the rewriting ruleset is $|\mathcal{U}| = 2W$, since each job could only switch its scheduling order with at most $W$ of its former and latter jobs respectively.

**Representation.** We represent each schedule as a directed acyclic graph (DAG), which describes the dependency among the schedule time of different jobs. Specifically, we denote each job $v_j$ as a node in the graph, and we add an additional node $v_0$ to represent the machine. If a job $v_j$ is scheduled at its arrival time $A_j$ (i.e., $B_j = A_j$), then we add a directed edge $\langle v_0, v_j \rangle$ in the graph. Otherwise, there must exist at least one job $v_{j'}$ such that $C_{j'} = B_j$ (i.e., job $j$ starts right after job $j'$). We add an edge $\langle v_{j'}, v_j \rangle$ for every such job $v_{j'}$ to the graph. Figure 2(b) shows the setting, and we defer the embedding and graph construction details to Appendix C.2.

**Model specification**. To encode the graphs, we extend the Child-Sum Tree-LSTM architecture in [46], which is similar to the DAG-structured LSTM in [53]. Similar to the expression simplification model, both the score predictor and the rule selector are fully connected neural networks, and we defer the model details to Appendix C.2.

## 5.3 Vehicle Routing Problem

In addition, we evaluate our approach on vehicle routing problems studied in [29, 35]. Specifically, we focus on the Capacitated VRP (CVRP), where a single vehicle with limited capacity needs to satisfy the resource demands of a set of customer nodes. To do so, we construct multiple routes starting and ending at the depot, i.e., node 0 in Figure 2(c), so that the resources delivered in each route do not exceed the vehicle capacity, while the total route length is minimized.

We represent each vehicle routing problem as a sequence of the nodes visited in the tour, and use a bi-directional LSTM to embed the routes. The ruleset is similar to the job scheduling, where each node can swap with another node in the route. The architectures of the score predictor and rule selector are similar to job scheduling. More details can be found in Appendix C.3.

# 6 Experiments

We present the evaluation results in this section. To calculate the inference time, we run all algorithms on the same server equipped with 2 Quadro GP100 GPUs and 80 CPU cores. Only 1 GPU is used when evaluating neural networks, and 4 CPU cores are used for search algorithms. We set the timeout of search algorithms to be 10 seconds per instance. All neural networks in our evaluation are implemented in PyTorch [38].

## 6.1 Expression Simplification

**Setup**. To construct the dataset, we first generate random pipelines using the generator in Halide, then extract expressions from them. We filter out those irreducible expressions, then split the rest into 8/1/1 for training/validation/test sets respectively. See Appendix A.1 for more details.

**Metrics**. We evaluate the following two metrics: (1) *Average expression length reduction*, which is the length reduced from the initial expression to the rewritten one, and the length is defined as the number of characters in the expression; (2) *Average tree size reduction*, which is the number of nodes decreased from the initial expression parse tree to the rewritten one.

**Baselines**. We examine the effectiveness of `NeuRewriter` against two kinds of baselines. The first kind of baselines are heuristic-based rewriting approaches, including `Halide-rule` (the rule-based Halide rewriter in Section 3) and `Heuristic-search`, which applies beam search to find the shortest rewriting with our ruleset at each step. Note that `NeuRewriter` does not use beam search.

In addition, we also compare our approach with Z3, a high-performance theorem prover developed by Microsoft Research [15]. Z3 provides two tactics to simplify the expressions: `Z3-simplify` performs some local transformation using its pre-defined rules, and `Z3-ctx-solver-simplify` traverses each sub-formula in the input expression and invokes the solver to find a simpler equivalent one to replace it. This search-based tactic is able to perform simplification not included in the Halide ruleset, and is generally better than the rule-based counterpart but with more computation. For `Z3-ctx-solver-simplify`, we set the timeout to be 10 seconds for each input expression.

**Results**. Figure 3a presents the main results. We can notice that the performance of `Z3-simplify` is worse than `Halide-rule`, because the ruleset included in this simplifier is more restricted than the Halide one, and in particular, it can not handle expressions with "max/min/select" operators. On the other hand, `NeuRewriter` outperforms both the rule-based rewriters and the heuristic search by a large margin. In particular, `NeuRewriter` could reduce the expression length and parse tree size by around $52\%$ and $59\%$ on average; compared to the rule-based rewriters, our model further reduces the average expression length and tree size by around $20\%$ and $15\%$ respectively. We observe that the main performance gain comes from learning to apply uphill rules appropriately in ways that are not included in the manually-designed templates. For example, consider the expression $5 \leq \max(\max(v0, 3) + 3, \max(v1, v2))$, which could be reduced to $True$ by expanding $\max(\max(v0, 3) + 3, \max(v1, v2))$ and $\max(v0, 3)$. Using a rule-based rewriter would require the need of specifying the pre-conditions recursively, which becomes prohibitive when the expressions become more complex. On the other hand, heuristic search may not be able to find the correct order of expanding the right hand size of the expression when more "min/max" are included, which would make the search less efficient.

Furthermore, `NeuRewriter` also outperforms `Z3-ctx-solver-simplify` in terms of both the result quality and the time efficiency, as shown in Figure 3a and Table 1a. Note that the implementation of Z3 is in C++ and highly optimized, while `NeuRewriter` is implemented in Python; meanwhile, `Z3-ctx-solver-simplify` could perform rewriting steps that are not included in the Halide ruleset. More results can be found in Appendix G.

**Generalization to longer expressions**. To measure the generalizability of our approach, we construct 4 subsets of the training set: $Train_{\leq 20}$, $Train_{\leq 30}$, $Train_{\leq 50}$ and $Train_{\leq 100}$, which only include expressions of length at most 20, 30, 50 and 100 in the full training set. We also build $Test_{>100}$, a subset of the full test set that only includes expressions of length larger than 100. The statistics of these datasets can be found in Appendix A.1.

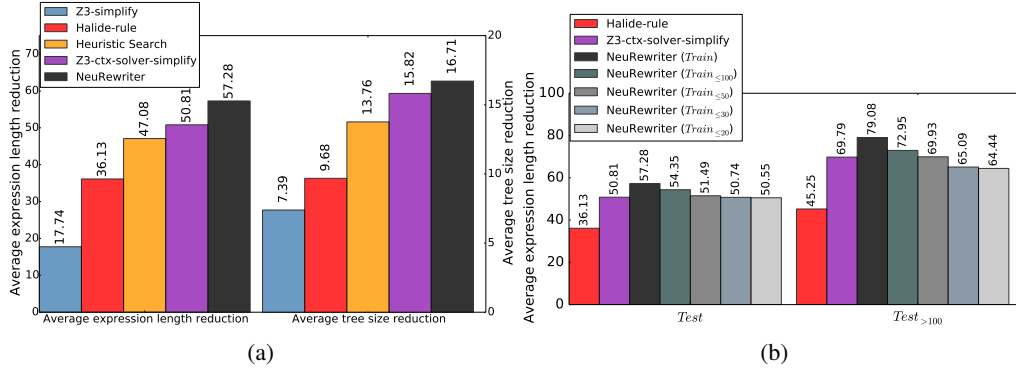

Figure 3: Experimental results of the expression simplification problem. In **(b)**, we train `NeuRewriter` on expressions of different lengths (described in the brackets).

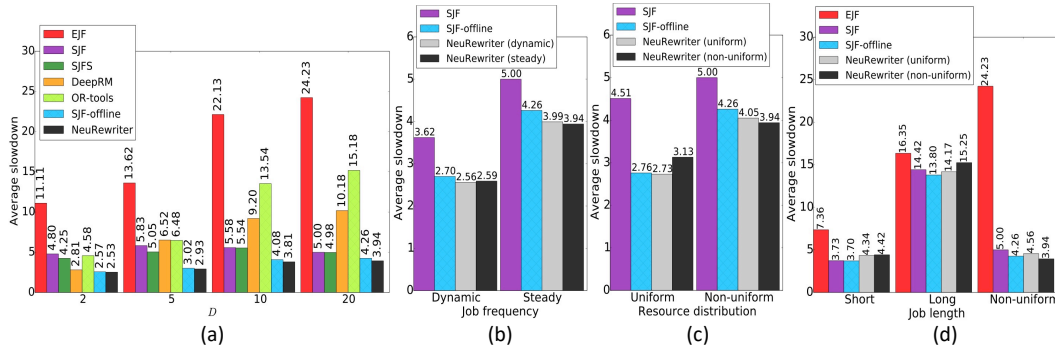

Figure 4: Experimental results of the job scheduling problem varying the following aspects: **(a)** the number of resource types $D$; **(b)** job frequency; **(c)** resource distribution; **(d)** job length. For `NeuRewriter`, we describe training job distributions in the brackets. Workloads in **(a)** are with steady job frequency, non-uniform resource distribution, and non-uniform job length. In **(b)**, **(c)** and **(d)**, $D = 20$. In **(b)** and **(c)**, we omit the comparison with some approaches because their results are significantly worse; for example, the average slowdown of EJF is $14.53$ on the dynamic job frequency, and $11.06$ on the uniform resource distribution. More results can be found in Appendix E.

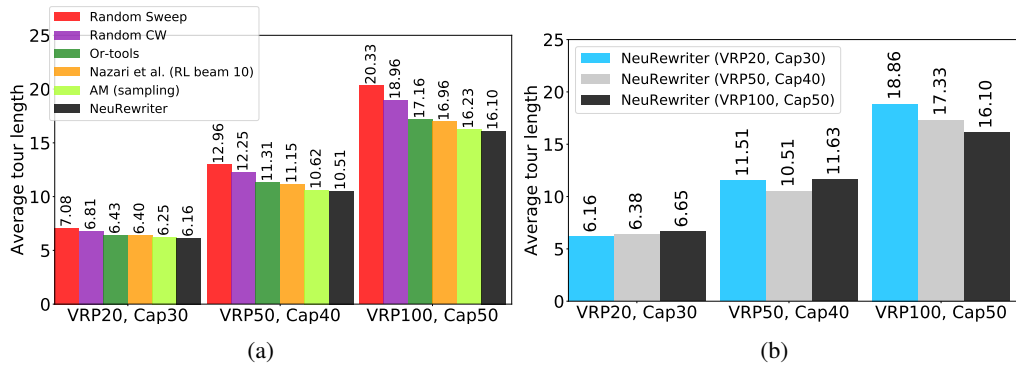

Figure 5: Experimental results of the vehicle routing problem with different number of customer nodes and vehicle capacity; e.g., VRP100, Cap50 means there are 100 customer nodes and the vehicle capacity is 50. **(a)** `NeuRewriter` outperforms multiple baselines and previous works [29, 35]. More results can be found in Appendix F. **(b)** We evaluate the generalization performance of `NeuRewriter` on problems from different distributions, and we describe the training problem distributions in the brackets.

We present the results of training our model on different datasets above in Figure 3b. Even trained on short expressions, `NeuRewriter` is still comparable with the Z3 solver. Thanks to local rewriting rules, our approach can generalize well even when operating on very different data distributions.

## 6.2   Job Scheduling Problem

**Setup**. We randomly generate $100K$ job sequences, and use $80K/10K/10K$ for training, validation and testing. Typically each job sequence includes $\sim 50$ jobs. We use an online setting where jobs arrive on the fly with a pending job queue of length $W = 10$. Unless stated otherwise, we generate initial schedules using *Earliest Job First* (`EJF`), which can be constructed with negligible overhead.

When the number of resource types $D = 2$, we follow the same setup as in [33]. The maximal job duration $T_{max} = 15$, and the latest job arrival time $A_{max} = 50$. With larger $D$, except changing the resource requirement of each job to include more resource types, other configurations stay the same.

**Metric**. Following `DeepRM` [33], we use the *average job slowdown* $\eta_j \equiv (C_j - A_j)/T_j \geq 1$ as our evaluation metric. Note that $\eta_j = 1$ means no slow down.

**Job properties.** To test the stability and generalizability of `NeuRewriter`, we change job properties (and their distributions): (1) *Number of resource types $D$*: larger $D$ leads to more complicated scheduling; (2) *Average job arrival rate*: the probability that a new job will arrive, *Steady job frequency* sets it to be $70\%$, and *Dynamic job frequency* means the job arrival rate changes randomly at each timestep; (3) *Resource distribution*: jobs might require different resources, where some are *uniform* (e.g., half-half for resource 1 and 2) while others are *non-uniform* (see Appendix A.2 for the detailed description); (4) *Job lengths*: *Uniform job length*: length of each job in the workload is either $[10, 15]$ (long) or $[1, 3]$ (short), and *Non-uniform job length*: workload has both short and long jobs. We show that `NeuRewriter` is fairly robust under different distributions. When trained on one distribution, it can generalize to others without performance collapse.

We compare `NeuRewriter` with three kinds of baselines.

**Baselines on Manually designed heuristics**: *Earliest Job First* (`EJF`) schedules each job in the increasing order of their arrival time. *Shortest Job First* (`SJF`) always allocates the shortest job in the pending job queue at each timestep, which is also used as a baseline in [33]. *Shortest First Search* (`SJFS`) searches over the shortest $k$ jobs to schedule at each timestep, and returns the optimal one. We find that other heuristic-based baselines used in [33] generally perform worse than `SJF`, especially with large $D$. Thus, we omit the comparison.

**Baselines on Neural network.** We compare with `DeepRM` [33], a neural network also trained with RL to construct a solution from scratch.

**Baselines on Offline planning**. To measure the optimality of these algorithms, we also take an offline setting, where the entire job sequence is available before scheduling. Note that this is equivalent to assuming an unbounded length of the pending job queue. With such additional knowledge, this setting provides a strong baseline. We tried two offline algorithms: (1) `SJF-offline`, which is a simple heuristic that schedules each job in the increasing order of its duration; and (2) Google OR-tools [19], which is a generic toolbox for combinatorial optimization. For OR-tools, we set the timeout to be 10 seconds per workload, but we find that it can not achieve a good performance even with a larger timeout, and we defer the discussion to Appendix E.

**Results on Scalability**. As shown in Figure 4a, `NeuRewriter` outperforms both heuristic algorithms and the baseline neural network `DeepRM`. In particular, while the performance of `DeepRM` and `NeuRewriter` are similar when $D = 2$, with larger $D$, `DeepRM` starts to perform worse than heuristic-based algorithms, which is consistent with our hypothesis that it becomes challenging to design a schedule from scratch when the environment becomes more complex. On the other hand, `NeuRewriter` could capture the bottleneck of an existing schedule that limits its efficiency, then progressively refine it to obtain a better one. In particular, our results are even better than offline algorithms that assume the knowledge of the entire job sequence, which further demonstrates the effectiveness of `NeuRewriter`. Meanwhile, we present the running time of OR-tools, `DeepRM` and `NeuRewriter` in Table 1b. We can observe that both `DeepRM` and `NeuRewriter` are much more time-efficient than OR-tools; on the other hand, the running time of `NeuRewriter` is comparable to `DeepRM`, while achieving much better results. More discussion can be found in Appendix E.

**Results on Robustness**. As shown in Figure 4, `NeuRewriter` excels in almost all different job distributions, except when the job lengths are uniform (short or long, Figure 4d), in which case

| | Time (s) |
|---|---|
| Z3-solver | 1.375 |
| NeuRewriter | 0.159 |

(a)

| | Time (s) |
|---|---|
| OR-tools | 10.0 |
| DeepRM | 0.020 |
| NeuRewriter | 0.037 |

(b)

| | VRP20 | VRP50 | VRP100 |
|---|---|---|---|
| OR-tools | 0.010 | 0.053 | 0.231 |
| Nazari et al. | 0.162 | 0.232 | 0.445 |
| AM | 0.036 | 0.168 | 0.720 |
| NeuRewriter | 0.133 | 0.211 | 0.398 |

(c)

Table 1: Average runtime (per instance) of different solvers (OR-tools [19] and the tactic `Z3-ctx-solver-simplify` of Z3 [15]) and RL-based approaches (`NeuRewriter`, DeepRM [33], Nazari et al. [35] and AM [29]) over the test set of: **(a)** expression simplification; **(b)** job scheduling; **(c)** vehicle routing.

existing methods/heuristics are sufficient. This shows that `NeuRewriter` can deal with complicated scenarios and is adaptive to different distributions.

**Results on Generalization.** Furthermore, `NeuRewriter` can also generalize to different distributions than those used in training, without substantial performance drop. This shows the power of local rewriting rules: using local context could yield more generalizable solutions.

## 6.3 Vehicle Routing Problem

**Setup and Baselines.** We follow the same training setup as [29, 35] by randomly generating vehicle routing problems with different number of customer nodes and vehicle capacity. We compare with two neural network approaches, i.e., AM [29] and Nazari et al. [35], and both of them train a neural network policy using reinforcement learning to construct the route from scratch. We also compare with OR-tools and several classic heuristics studied in [35].

**Results.** We first demonstrate our main results in Figure 5a, where we include the variant of each baseline that performs the best, and defer more results to Appendix F. Note that the initial routes generated for `NeuRewriter` are even worse than the classic heuristics; however, starting from such sub-optimal solutions, `NeuRewriter` is still able to iteratively improve the solutions and outperforms all the baseline approaches on different problem distributions. In addition, for VRP20 problems, we can compute the exact optimal solutions, which provides an average tour length of 6.10. We observe that the result of `NeuRewriter` (i.e., 6.16) is the closest to this lower bound, which also demonstrates that `NeuRewriter` is able to find solutions with better quality.

We also compare the runtime of the most competitive approaches in Table 1c. Note that the OR-Tools solver for vehicle routing problems is highly tuned and implemented in C++, while the RL-based approaches in comparison are implemented in Python. Meanwhile, following [35], to report the runtime of RL models, we decode a single instance at a time, thus there is potential room for speed improvement by decoding multiple instances per batch. Nevertheless, we can still observe that `NeuRewriter` achieves a better balance between the result quality and the time efficiency, especially with a larger problem scale.

**Results on Generalization.** Furthermore, in Figure 5b, we show that `NeuRewriter` can generalize to different problem distributions than training ones. In particular, they still exceed the performance of the classic heuristics, and are sometimes comparable or even better than the OR-tools. More discussion can be found in Appendix F.

## 7 Conclusion

In this work, we propose to formulate optimization as a rewriting problem, and solve the problem by iteratively rewriting an existing solution towards the optimum. We utilize deep reinforcement learning to train our neural rewriter. In our evaluation, we demonstrate the effectiveness of our neural rewriter on multiple domains, where our model outperforms both heuristic-based algorithms and baseline deep neural networks that generate an entire solution directly.

Meanwhile, we observe that since our approach is based on local rewriting, it could become time-consuming when large changes are needed in each iteration of rewriting. In extreme cases where each rewriting step needs to change the global structure, starting from scratch becomes preferrable. We consider improving the efficiency of our rewriting approach and extending it to more complicated scenarios as future work.

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
