[Supplementary Material · supp.pdf]

```
       <Expr>    ::=  <AlgExpr> | <BoolExpr>
   <BoolExpr>    ::=  <AlgExpr> < <AlgExpr>
                   |  <AlgExpr> <= <AlgExpr>
                   |  <AlgExpr> == <AlgExpr>
                   |  (! <BoolExpr>)
                   |  (<BoolExpr> && <BoolExpr>)
                   |  (<BoolExpr> || <BoolExpr>)
    <AlgExpr>    ::=  <Term>
                   |  (<AlgExpr> + <Term>)
                   |  (<AlgExpr> - <Term>)
                   |  (<AlgExpr> * <Term>)
                   |  (<AlgExpr> / <Term>)
                   |  (<AlgExpr> % <Term>)
       <Term>    ::=  <Var> | <Const>
                   |  max(<AlgExpr>, <AlgExpr>)
                   |  min(<AlgExpr>, <AlgExpr>)
                   |  select(<BoolExpr>, <AlgExpr>, <AlgExpr>)
```

Figure 6: Grammar of the Halide expressions in our evaluation. "select $(c, e1, e2)$" means that when the condition $c$ is satisfied, this term is equal to $e1$, otherwise is equal to $e2$. In our dataset, all constants are integers ranging in $[-1024, 1024]$, and variables are from the set $\{v0, v1, ..., v12\}$.

| Number of expressions in the dataset | Length of expressions | Size of expression parse trees |
|---|---|---|
| Total: 1.36M | Average: 106.84 | Average: 27.39 |
| Train/Val/Test: 1.09M/136K/136K | Min/Max: 10/579 | Min/Max:3/100 |
| $Train_{\leq 20}$: 17K | Average: 16.76 | Average: 4.66 |
| $Train_{\leq 30}$: 48K | Average: 22.91 | Average: 6.43 |
| $Train_{\leq 50}$: 170K | Average: 35.62 | Average: 10.18 |
| $Train_{\leq 100}$: 588K | Average: 63.49 | Average: 18.72 |
| $Test_{>100}$: 53K | Average: 142.22 | Average: 42.20 |

Table 2: Statistics of the dataset for expression simplification.

## A   More Details of the Dataset

### A.1   Expression Simplification

Figure 6 presents the grammar of Halide expressions in our evaluation. We use the random pipeline generator in the Halide repository to build the dataset [3]. Table 2 presents the statistics of the datasets.

### A.2   Job Scheduling

**Description of different resource distributions.**    For each job $j$, we define *dominant resources* $d_{\mathrm{dom}}$ as the resources with $0.5 \leq \rho_{jd_{\mathrm{dom}}} \leq 1$, and *auxiliary resources* $d_{\mathrm{aux}}$ as those with $0.1 \leq \rho_{jd_{\mathrm{aux}}} \leq 0.2$. We refer to a job with both dominant and auxiliary resources as a job with *non-uniform resources*. We also evaluate on workloads including only jobs with *uniform resources*, where each job only includes either dominant resources or auxiliary resources.

### A.3   Vehicle Routing

Our data generation follows the setup in [35, 29]. The positions of the depot and customer nodes are uniformly randomly sampled from the unit square $[0, 1] \times [0, 1]$. Each node is denoted as $v_j = ((x_j, y_j), \delta_j)$, where $(x_j, y_j)$ is the position, and $\delta_j$ is the resource demand. We set $\delta_0 = 0$ for the depot (i.e., node 0), and $\delta_j \in \{1, 2, ..., 9\}$ for customer nodes (i.e., $j > 0$).

Figure 7: An example of the rewriting process for Halide expressions. The initial expression is $5 \leq max(v0, 3) + 3$, which could be reduced to 1, i.e., $True$.

---

**Algorithm 1** Algorithm of a Single Rewriting Step for Job Scheduling Problem

---

1: **function** REWRITE($v_j, v_{j'}, s_t$)
2:     **if** $C_{j'} < A_j$ or $C_{j'} == B_j$ **then**
3:         **return** $s_t$
4:     **end if**
5:     **if** $j' \neq 0$ **then** $B'_j = C_{j'}$ **else** $B'_j = A_j$ **fi**
6:     $C'_j = B'_j + T_j$
7:
8:     //Resolve potential resource occupation overflow within $[B'_j, C'_j]$
9:     $J =$ all jobs in $s_t$ except $v_j$ that are scheduled within $[B'_j, C'_j]$
10:     Sort $J$ in the topological order
11:     **for** $v_i \in J$ **do**
12:         $B'_i =$ the earliest time that job $v_i$ can be scheduled
13:         $C'_i = B'_i + T_i$
14:     **end for**
15:     For $v_i \notin J$, $B'_i = B_i$, $C'_i = C_i$
16:     $s_{t+1} = \{(B'_i, C'_i)\}$
17:     **return** $s_{t+1}$
18: **end function**

---

# B  More Details on the Rewriting Ruleset

## B.1  More Details for Expression Simplification Problem

The ruleset implemented in the Halide rule-based rewriter can be found in their public repository [4].

**More discussions about the uphill rules.**   A commonly used type of uphill rules is "min/max" expansion, e.g., $\min(a, b) < c \rightarrow a < c || b < c$. Dozens of templates in the ruleset of the Halide rewriter are describing conditions when a "min/max" expression could be simplified. Notice that although applying this rewriting rule has no benefit in most cases, since it will increase the expression length, it is necessary to include it in the ruleset, because when either $a < c$ or $b < c$ is always true, expanding the "min" term could reduce the entire expression to a tautology, which ends up simplifying the entire expression. Figure 7 shows an example of the rewriting process using uphill rules properly.

## B.2  More Details for Job Scheduling Problem

Algorithm 1 describes a single rewriting step for job scheduling problem.

Figure 8: An example to illustrate the job embedding approach for the job scheduling problem.

Figure 9: An example to illustrate two possible job schedules on a single machine and their corresponding graph representations. Node 0 was added to represent the start of the scheduling process. For multiple machines, multiple node 0 will be added.

## C   More Details on Model Architectures

### C.1   Model Details for Expression Simplification

**Input embedding.**   Notice that in this problem, each non-terminal has at most 3 children. Thus, let $x$ be the embedding of a non-terminal, $(h_L, c_L), (h_M, c_M), (h_R, c_R)$ be the LSTM states maintained by its children nodes, the LSTM state of the non-terminal node is computed as

$$(h, c) = \text{LSTM}(([h_L; h_M; h_R], [c_L; c_M; c_R]), x) \tag{4}$$

Where $[a; b]$ denotes the concatenation of vectors $a$ and $b$. For non-terminals with less than 3 children, the corresponding LSTM states are set to be zero. We use $d$ to represent the size of $h$ and $c$, i.e., the hidden size of the LSTM.

**Input representation.**   For each sub-tree $\omega_i$, its input to both the score predictor and the rule-picking policy is represented as a $2d$-dimensional vector $[h_0; h_i]$, where $h_0$ is the embedding of the root node encoding the entire tree. The reason why we include $h_0$ in the input is that looking at the sub-tree itself is sometimes insufficient to determine whether it is beneficial to perform the rewriting. For example, consider the expression $max(a, b) + 2 < a + 2$, by looking at the sub-expression $max(a, b) + 2$ itself, it does not seem necessary to rewrite it as $max(a + 2, b + 2)$. However, given the entire expression, we can observe that this rewriting is an important step towards the simplification, since the resulted expression $max(a + 2, b + 2) < a + 2$ could be reduced to $False$. We have tried other approaches of combining the parent information into the input, but we find that including the embedding of the entire tree is the most efficient way.

**Score predictor.**   The score predictor is an $L_P$-layer fully connected network with a hidden size of $N_P$. For each sub-tree $\omega_i$, its input to the score predictor is represented as a $2d$-dimensional vector $[h_0; h_i]$, where $h_0$ embeds the entire tree.

**Rule selector.**   The rule selector is an $L_S$-layer fully connected network with the hidden size $N_S$, and its input format is the same as the score predictor. A $|\mathcal{U}|$-dimensional softmax layer is used as the output layer.

### C.2   More Details for Job Scheduling Problem

**Job embedding.**   We embed each job into a $(D \times (T_{max} + 1) + 1)$-dimensional vector $e_j$, where $T_{max}$ is the maximal duration of a job. This vector encodes the information of the job attributes and the machine status during its execution. We describe the details of job embedding as follows. Consider

a job $v_j = (\boldsymbol{\rho}_j, A_j, T_j)$. We denote the amount of resources occupied by all jobs at each timestep $t$ as $\boldsymbol{\rho}'_t = (\rho'_{t1}, \rho'_{t2}, ..., \rho'_{tD})$. Each job $v_j$ is represented as a $(D \times (T_{max} + 1) + 1)$-dimensional vector, where the first $D$ dimensions of the vector are $\boldsymbol{\rho}_j$, representing its resource requirement. The following $D \times T_j$ dimensions of the vector are the concatenation of $\boldsymbol{\rho}'_{B_j}, \boldsymbol{\rho}'_{B_j+1}, ..., \boldsymbol{\rho}'_{B_j+T_j-1}$, which describes the machine usage during the execution of the job $v_j$. When $T_j < T_{max}$, the following $D \times (T_{max} - T_j)$ dimensions are zero. The last dimension of the embedding vector is the slowdown of the job in the current schedule. We denote the embedding of each job $v_j$ as $e_j$. The embedding of the machine (i.e., $v_0$) is a zero vector $e_0 = \mathbf{0}$. Figure 8 shows an example of our job embedding approach, and Figure 9 illustrates an example of the graph construction.

**Model specification**. To encode the graphs, we extend the Child-Sum Tree-LSTM architecture in [46], which is similar to the DAG-structured LSTM in [53]. Specifically, for a job $v_j$, suppose $(h_1, c_1), (h_2, c_2), ..., (h_p, c_p)$ are the LSTM states of all parents of $v_j$, then its LSTM state is:

$$(h, c) = \text{LSTM}((\sum_{i=1}^{p} h_i, \sum_{i=1}^{p} c_i), e_j) \tag{5}$$

For each node, the $d$-dimensional hidden state $h$ is used as the embedding for other two components.

**Score predictor.** This component is an $L_P$-layer fully connected neural network with a hidden size of $N_P$, and the input to the predictor of job $v_j$ is $h_j$.

**Rule selector.** The rewriting rules are equivalent to moving the current job $v_j$ to be a child of another job $v_{j'}$ or $v_0$ in the graph, which means allocating job $v_j$ after job $v_{j'}$ finishes or at its arrival time $A_j$. Thus, the input to the rule selector not only includes $h_j$, but also $h_{j'}$ of all other $v_{j'}$ that could be used for rewriting. The rule selector has two modules. The first module is an $L_S$-layer fully connected neural network with a hidden size of $N_S$. For each job $v_j$, let $N_j$ be the number of jobs that could be the parent of $v_j$, and $\{v_{j'_k}\}$ denotes the set of such jobs. For each $v_{j'_k}$, the input is $[h_j; h_{j'_k}]$, and this module computes a $d$-dimensional vector $h'_k$ to encode such a pair of jobs. The second module of the rule selector is another $L_S$-layer fully connected neural network with a hidden size of $N_S$. For this module, the input is a $(|\mathcal{U}| \times d)$-dimensional vector $[h'_1; h'_2; ...; h'_{|\mathcal{U}|}]$, where $|\mathcal{U}| \models 2W$. When $N_j < |\mathcal{U}|$, $h'_{N_j+1}, h'_{N_j+2}, ..., h'_{|\mathcal{U}|}$ are set to be zero. The output layer of this module is a $|\mathcal{U}|$-dimensional softmax layer, which predicts the probability of each different move of $v_j$.

### C.3 More Details for Vehicle Routing Problem

**Node embedding.** We embed each node into a 7-dimensional vector $e_j$. This vector encodes the information of the node position, node resource demand, and the current status of the vehicle. We describe the details of node embedding as follows. Consider a node $v_j = ((x_j, y_j), \delta_j)$, where $(x_j, y_j)$ is the position, and $\delta_j$ is the resource demand. We set $\delta_0 = 0$ for the depot (i.e., node 0). Denote $Cap$ as the vehicle capacity. The first three dimensions of $e_j$ are $x_j$, $y_j$, and $\delta_j/Cap$. The next three dimensions of $e_j$ are the coordinates of the node visited at the previous step (set as the depot position for the first visited node) and the Euclidean distance between $v_j$ and the previous node. The last dimension is the amount of remaining resources carried by the vehicle at the current step, which is also normalized by the vehicle capacity.

**Score predictor.** This component is an $L_P$-layer fully connected neural network with a hidden size of $N_P$, and the input to the predictor of the node $v_j$ is $h_j$, where $h_j$ is the output of the bi-directional LSTM used to encode each node in the route.

**Rule selector.** The rewriting rules are equivalent to moving a node in the route $v_j$ after another node $v_{j'}$, similar to the job scheduling setting. However, different from job scheduling, the number of such nodes $v_{j'}$ varies among different problems. Thus, we train an attention module to select $v_{j'}$, with a similar design to the pointer network [50].

---

**Algorithm 2** Forward Pass Algorithm for the Neural Rewriter during Training

---

**Require:** initial state $s_0$, hyper-parameters $\epsilon, p_c, T_{iter}, T_\omega, T_u$

1: **for** $t = 0 \to T_{iter} - 1$ **do**
2:      **for** $i = 1 \to T_\omega$ **do**
3:          Sample $\omega_t \sim \pi_\omega(\omega_t|s_t; \theta)$, where $\omega_t \in \Omega(s_t)$, $\pi_\omega(\omega_t|s_t; \theta) = \frac{\exp(Q(s_t, \omega_t; \theta))}{\sum_{\omega_t} \exp(Q(s_t, \omega_t; \theta))}$
4:          **if** $Q(s_t, \omega_t; \theta) < \epsilon$ **then**
5:              Re-sample $\omega_t' \sim \pi_\omega(\omega_t'|s_t; \theta)$ with a probability of $1 - p_c$
6:              **if** Re-sampling is not performed **then break fi**
7:          **else**
8:              **break**
9:          **end if**
10:      **end for**
11:      **for** $i = 1 \to T_u$ **do**
12:          Sample $u_t \sim \pi_u(u_t|s_t[\omega_t]; \phi)$
13:          **if** $u_t$ can be applied to $s_t[\omega_t]$ **then break fi**
14:      **end for**
15:      **if** $u_t$ does not applied to $s_t[\omega_t]$ **then break fi**
16:      $s_{t+1} = f(s_t, \omega_t, u_t)$
17: **end for**

---

### C.4    Model hyper-parameters

For both the expression simplification and job scheduling tasks, $L_S = L_P = 1$. For the vehicle routing task, $L_S = L_P = 2$. For all the three tasks, $N_S = N_P = 256, d = 512$.

## D    More Details on Training

Algorithm 2 presents the details of the forward pass during training. The forward pass during evaluation is similar, except that we compute $\omega_t$ and $u_t$ as $\omega_t = \arg\max_\omega \pi_\omega(\omega|s_t; \theta)$ and $u_t = \arg\max_u(\pi_u(u|s_t[\omega_t]; \phi))$, and the inference immediately terminates when $Q(s_t, \omega_t; \theta) < \epsilon$ or $u_t$ does not apply.

**Hyper-parameters.** For all tasks in our evaluation, in Algorithm 2, $\epsilon = 0.0, T_\omega = 10, T_u = 10$. $T_{iter} = 50$ for both the expression simplification and the job scheduling tasks. For the vehicle routing task, we set $T_{iter} = 200$, because we find that applying 50 rewriting steps could be insufficient for finding a competitive solution, especially when the number of customer nodes is large. For all tasks in our evaluation, $p_c$ is initialized with $0.5$, and is decayed by $0.8$ for every 1000 timesteps until $p_c = 0.01$, where it is not decayed anymore. In the training loss function, $\alpha = 10.0$. The decay factor for the cumulative reward is $\gamma = 0.9$. The initial learning rate is $1e - 4$, and is decayed by $0.9$ for every 1000 timesteps. Batch size is 128. Gradients with $L_2$ norm larger than $5.0$ are scaled down to have the norm of $5.0$. The model is trained using Adam optimizer. All weights are initialized uniformly randomly in $[-0.1, 0.1]$.

## E    More Results for Job Scheduling Problem

We observe that while OR-tools is a high-performance solver for generic combinatorial optimization problems, it is less effective than both heuristic-based scheduling algorithms and neural network approaches on our job scheduling problem, especially with more resource types. After looking into the schedules computed by OR-tools, we find that they often prioritize long jobs over short jobs, while swapping the scheduling order between them would clearly decrease the job waiting time. On the other hand, both our neural rewriter and heuristic algorithms based on the job length would usually schedule short jobs very soon after their arrival, which results in better schedules.

Table 3 and 4 present the results of ablation study on job frequency and resource distribution respectively.

|                             | Dynamic Job Frequency | Steady Job Frequency |
|-----------------------------|:---------------------:|:--------------------:|
| Earliest Job First (EJF)    | 14.53                 | 24.23                |
| Shortest Job First (SJF)    | 3.62                  | 5.00                 |
| SJF-offline                 | 2.70                  | 4.26                 |
| NeuRewriter (dynamic)       | **2.56**              | 3.99                 |
| NeuRewriter (steady)        | 2.59                  | **3.94**             |

Table 3: Experimental results of the job scheduling problem with different distribution of job frequency.

|                              | Uniform Job Resources | Non-uniform Job Resources |
|------------------------------|:---------------------:|:-------------------------:|
| Earliest Job First (EJF)     | 11.06                 | 24.23                     |
| Shortest Job First (SJF)     | 4.51                  | 5.00                      |
| SJF-offline                  | 2.76                  | 4.26                      |
| NeuRewriter (uniform)        | **2.73**              | 4.05                      |
| NeuRewriter (non-uniform)    | 3.13                  | **3.94**                  |

Table 4: Experimental results of the job scheduling problem with different distribution of job resources.

| Initial average slowdown    | $\leq 10$ | $10-25$ | $> 25$ |
|-----------------------------|:---------:|:-------:|:------:|
| Final average slowdown      | 3.88      | 3.90    | 4.06   |

| Earliest Job First (EJF)    | 24.23 |
|-----------------------------|:-----:|
| Shortest Job First (SJF)    | 5.00  |
| Shortest First Search (SJFS)| 4.98  |
| DeepRM                      | 10.18 |
| OR-tools                    | 15.18 |
| SJF-offline                 | 4.26  |
| NeuRewriter                 | **3.94** |

Table 5: Experimental results of the job scheduling problem using initial schedules with different average slowdown. The number of resource types $D = 20$.

To examine how the initial schedules affect the final results, besides earliest-job-first schedules, we also evaluate initial schedules with different average slowdown. Specifically, for each job sequence, we generate different initial schedules by randomly allocating one job at a time.

In Table 5, we present the results with $D = 20$ types of resources. For each job sequence, we randomly generate 10 different initial schedules. We can observe that although the effectiveness of initial schedules affects the final schedules, the performance is still consistently better than other baseline approaches, which demonstrates that our neural rewriter is able to substantially improve the initial solution regardless of its quality.

## F   More Discussion of the Evaluation on Vehicle Routing Problem

We generate the initial routes for `NeuRewriter` in the following way: starting from the depot, at each timestep, the vehicle visits the nearest node that is either: (1) a customer node that has not been visited yet, and its resource demand can be satisfied; or (2) the depot node, and the resources carried by the vehicle is less than its capacity. See Figure 10 for examples of the initial solutions. In this way, the average tour length is 7.74 for VRP20, 13.47 for VRP50, and 20.36 for VRP100. Note that these results are even worse than the classic heuristics compared in Table 6.

Table 6 presents more results for vehicle routing problems, and Figure 10 shows an example of the rewriting steps performed by `NeuRewriter`.

For generalization results, note that after training on VRP50, `NeuRewriter` achieves an average tour length of 17.33 on VRP100 (See Figure 5b in the mainbody of the paper). This is better than 18.00 reported in [35], suggesting that our approach could adapt better to different problem distributions.

| Model | VRP20, Cap30 | VRP50, Cap40 | VRP100, Cap50 |
|---|---|---|---|
| NeuRewriter | **6.16** | **10.51** | **16.10** |
| AM-Greedy | 6.40 | 10.98 | 16.80 |
| AM-Sampling | 6.25 | 10.62 | 16.23 |
| Nazari et al. (RL-Greedy) | 6.59 | 11.39 | 17.23 |
| Nazari et al. (RL-BS(5)) | 6.45 | 11.22 | 17.04 |
| Nazari et al. (RL-BS(10)) | 6.40 | 11.15 | 16.96 |
| CW-Greedy | 7.22 | 12.85 | 19.72 |
| CW-Rnd(5,5) | 6.89 | 12.35 | 19.09 |
| CW-Rnd(10,10) | 6.81 | 12.25 | 18.96 |
| SW-Basic | 7.59 | 13.61 | 21.01 |
| SW-Rnd(5) | 7.17 | 13.09 | 20.47 |
| SW-Rnd(10) | 7.08 | 12.96 | 20.33 |
| OR-Tools | 6.43 | 11.31 | 17.16 |
| Gurobi (optimal) | 6.10 | - | - |

Table 6: Experimental results of the vehicle routing problems.

(a) Step 0.

(b) Step 1.

(c) Steps 2-5.

(d) Step 6.

Figure 10: An example of the rewriting steps for a VRP20 problem. The square is the depot, and circles are customer nodes. The customer node sizes are proportional to their resource demands. At each stage, red edges are to be rewritten at the next step, and green edges are rewritten ones. The tour length of the initial route is 7.31, and the final tour length after rewriting is 5.98.

# G   More Results for Expression Simplification

In Figures 11 and 12, we present some success cases of expression simplification, where we can simplify better than both the Halide rule-based rewriter and the Z3 solver.

(a) Step 0.      (b) Step 1.

(c) Step 2.      (d) Step 3.

(e) Step 4.

Figure 11: The rewriting process that simplifies the expression $((v0 - v1 + 18)/35 * 35 + 35) \leq v0 - v1 + 119$ to $34 \leq (v0 - v1 + 13)\%35$.

(a) Step 0.      (b) Step 1.

(c) Step 2.      (d) Step 3.

(e) Step 4.      (f) Step 5.

(g) Step 6.      (h) Step 7.

(i) Step 8.      (j) Step 9.

(k) Step 10.

Figure 12: The rewriting process that simplifies the expression $((v0 - v1 + 12)/137 * 137 + 137) \leq min((v0 - v1 + 149)/137 * 137, v0 - v1 + 13)$ to $136 \leq (v0 - v1 + 12)\%137$.

## Footnotes

[3]https://github.com/halide/Halide/tree/new_autoschedule_with_new_simplifier/apps/random_pipeline.

[4] https://github.com/halide/Halide.