[Reviews · NeurIPS 2019]

Reviewer 1



The paper proposes a new end-to-end method that can solve a range of combinatorial optimization (CP) problems. Although such RL approaches have been proposed before, a novel two-stage approach is devised in this paper and extensive experiments demonstrate its effectiveness. Weakness: 1. The improvement of the proposed method over existing RL method is not impressive. 2. Compared to OR tools and RL baselines, the time and computational cost should be reported in detail to fairly compare different methods. Comment after feedback: The authors have addressed the concerns of running time. Since the applying RL in combinatorial optimization is not new, the lack of comparisons between existing RL methods makes it less convincing. Reinforcement Learning for Solving the Vehicle Routing Problem, Mohammadreza Nazari. ATTENTION, LEARN TO SOLVE ROUTING PROBLEMS!, Max Welling. Exact Combinatorial Optimization with Graph Convolutional Neural Networks, Maxime Gasse. Learning Combinatorial Optimization Algorithms over Graphs, Le Song.

Reviewer 2



Originality: The work proposes an interesting approach with a factorized policy to perform RL by performing iterative improvement over the current solution until convergence is achieved. Using RL for combinatorial optimization is not new, even though the specific idea that the authors propose differs from previous works that I am aware of. Quality: I believe the work is technically sound. It uses standard policy gradient theorems and the well-known actor-critic framework. Clarity: The paper is clearly written with sufficient experimental details. On the other hand, I would like to see more about connection to local search and some further improvements (see point 5 below). Significance: Given that (1) solving NP-hard combinatorial problems is a very important task and (2) the experimental results outperform standard solvers, I think that the current work is fairly significant due to its practicability. UPDATE I have read both the rebuttal and the other reviews. I appreciate the authors' feedback, particularly with regard to adding more information on the connections to local search, and the wider applicability of this work. For this reason, I am willing to increase my score to 7. I would just encourage the authors to ensure that, if accepted, the camera-ready will clarify all points raised in the rebuttal.

Reviewer 3



After rebuttal: The discussion of the method applicability in the rebuttal is convinced for me. I upgrade my score to 7. This paper proposes a learning-based approach for combinatorial optimization problems. Starting from an initial complete solution of the problem, several local rewriting updates are applied to the solution iteratively. In each rewriting step, a local region and an updating rule are picked to update the solution and two networks are trained by reinforcement learning to pick local regions and updating rules. Experiments on expression simplification, job scheduling and vehicle routing problems are conducted to validate the effectiveness of the proposed approach. The main concern of this paper is the applicable range of the proposed method is unclear. I think the local rewriting approach stands under several assumptions, the complete and legal solution is easily initialized, the reasonable local regions and updating rules can be manually defined, the rewards are densely distributed for solutions. It seems this approach may not work in the problem where we need to find a specific target and therefore the state transition is important. For example, in the problem of theorem proving, I don't think it could work to rewrite the proofs directly if we think proofs are the expected solutions, as stated in line 63. Also, how to design local regions and updating rules may be important and ambiguous in specific problems. I agree that local rewriting should be applicable to a wide range of problems, but I think this range and underlying assumptions should be formally and explicitly clarified in the paper. For example, to demonstrate the targeted problems in the introduction. It could help to clarify the contributions of this work and explain the choices of tasks in experiments.

[Author Response · NeurIPS 2019]

| Approach | Time (s) |
|---|---|
| Z3 (solver) | 1.375 |
| NeuRewriter | 0.159 |

(a)

| Approach | Time (s) |
|---|---|
| OR-tools | 10.0 |
| DeepRM | 0.020 |
| NeuRewriter | 0.037 |

(b)

| | VRP20 | VRP50 | VRP100 |
|---|---|---|---|
| OR-tools | 0.010 | 0.053 | 0.231 |
| Nazari et al. | 0.162 | 0.232 | 0.445 |
| AM | 0.036 | 0.168 | 0.720 |
| NeuRewriter | 0.133 | 0.211 | 0.398 |

(c)

Table 1: Average running time over the test set of: **(a)** expression simplification; **(b)** job scheduling; **(c)** vehicle routing.

We thank all reviewers for valuable and encouraging comments!

**Main Contributions**. We demonstrate that a neural network-based policy can be trained with RL to replace laborious
work of tuning heuristics, and in specific problems (expression simplification, job scheduling and vehicle routing),
outperforms extensive baselines including (1) common heuristics; (2) heuristics-guided search; (3) software (Halide
and Z3 in Expression simplification, OR-tools in Job scheduling and Vehicle routing); and (4) previous deep models.
Our policy is factorized into region-picking and rule-picking components to decide where and which rewriting rule
to apply. Meanwhile, our method demonstrates strong generalizability when the model is evaluated on different data
distributions than the training one, suggesting the power of leveraging local structures.

**Method Applicability and Limitation**. We agree that our approach demonstrates its greatest strengths when the
problems have well-behaved local structures in the search space. We have briefly mentioned the assumptions (e.g., Line
67-77). Based on suggestions from R2 and R3, we will revise our paper to provide a clearer and more formal discussion
of the applicable range of problems and the underlying assumptions of our approach.

Note that our method can be applied even if a feasible solution is hard to find: we can start with an infeasible solution,
then move towards a feasible one by using the reduction of infeasibility (e.g., more constraints are satisfied) as the
reward in RL. The method can also be applied to situations with sparse reward, since $Q(s_t, \omega_t; \theta)$ is trained supervisedly
with the accumulated reward in each rewriting trajectory. The learned policy is also not blindly driven by short-term
reward. For example, in expression simplification, the model learns to properly apply a few uphill rules to make the
expression longer (and receive negative rewards), before reaching a solution with a high reward (Line 231-238).

Although the proposed method is mainly evaluated on combinatorial optimization, it can be extended to other rewriting
problems, e.g., equational theorem proving solvable by rewriting-based systems [1, 3], and we leave it as future work.

**Reviewer 1.** See Table 1 for the average running time per problem instance. We have described the details of time
calculation in our submission (Line 200-204, 223 and 283-285). Note that the implementation of Z3 and OR-tools
are in C++, while NeuRewriter and RL baselines are in Python. Still, we can observe that our approach achieves a
better balance between the time-efficiency and the result quality. For expression simplification and job scheduling,
NeuRewriter is even more time-efficient than Z3 and OR-tools. Meanwhile, the time-efficiency of NeuRewriter is
comparable or better than RL baselines, while achieving better results. We will provide more discussion in our revision.

**Reviewer 2.** We agree that at a high level, our framework is closely connected with the local search pipeline. Specifically,
we can leverage our learned RL policy to guide the local search, i.e., to decide which neighbor solution to move to.
We have compared with several approaches using local search guided by manually designed heuristics, e.g., Heuristic
Search for expression simplification (Line 215-216), Random CW for vehicle routing (Figure 5 in page 6), and
softwares supporting more advanced local search algorithms (Z3 and OR-tools), and we demonstrate that our approach
outperforms these baselines in our tasks. We will further highlight this connection in our revision.

The region-picker $\pi_\omega$ is parameterized by a $Q$-function and is similar in spirit to soft-$Q$ learning [2]. We learn $Q$
function directly from the final rewards of the trajectory in a supervised manner, and thus avoid boot-strapping which
could cause sample insufficiency and instability in training. We have also tried the bootstrapping estimation (e.g., $n$-step
TD), and the results are not better.

**Reviewer 3.** See above for applicability of the proposed method. We will definitely release the code in the final version.

# References

[1] L. Bachmair and H. Ganzinger. Rewrite-based equational theorem proving with selection and simplification. *Journal of Logic*
*and Computation*, 4(3):217–247, 1994.

[2] T. Haarnoja, H. Tang, P. Abbeel, and S. Levine. Reinforcement learning with deep energy-based policies. In *ICML*, pages
1352–1361. JMLR. org, 2017.

[3] J. Hsiang, H. Kirchner, P. Lescanne, and M. Rusinowitch. The term rewriting approach to automated theorem proving. *The*
*Journal of Logic Programming*, 14(1-2):71–99, 1992.


[Meta-Review · NeurIPS 2019]

The reviewers liked the paper and were further convinced by the response. Well done! Please take their suggestions into account when preparing the final version of your paper.